# “It’s Like Living with a Sassy Teenager!”: A Mixed-Methods Analysis of Owners’ Comments about Dogs between the Ages of 12 Weeks and 2 Years

**DOI:** 10.3390/ani13111863

**Published:** 2023-06-03

**Authors:** Sara C. Owczarczak-Garstecka, Rosa E. P. Da Costa, Naomi D. Harvey, Kassandra Giragosian, Rachel H. Kinsman, Rachel A. Casey, Séverine Tasker, Jane K. Murray

**Affiliations:** 1Dogs Trust, Canine Behaviour and Research Department, 17 Wakely Street, London EC1V 7RQ, UK; 2Linnaeus Veterinary Limited, 1011 Stratford Road, Solihull B90 4BN, UK; 3Bristol Veterinary School, University of Bristol, Bristol BS40 5DU, UK

**Keywords:** adolescence, dog behaviour, human–animal interactions, open-ended questions, longitudinal study

## Abstract

**Simple Summary:**

Owners’ understanding of dog behaviour influences dog welfare. This study explored owners’ experiences and perceptions of dog behaviour. Data came from an ongoing UK/ROI study of dogs. Survey questions when dogs were 12/16 weeks (data combined), 6, 12, 18 and 24 months were analysed. Data were explored with two approaches: (1) qualitative thematic analysis and (2) quantitative text analysis. Responses to ‘other information’ questions and those regarding owner-reported problem behaviours were explored to understand owners’ experiences/understanding of dog behaviour (1). Responses to the ‘other information’ questions were evaluated to understand how sentiment in the text and in word use changes over time (2). The proportion of positive: negative sentiments increased with the dog’s age. At the first time point, ‘bite’ was the most common word, later replaced by words related to ‘love’. Owners referred to the ‘dog’s biology’, ‘personality/deliberate action’ and ‘external influences’ when explaining dogs’ behaviour. Problematic behaviours of young dogs were seen as ‘mischievous’, unintentional and context-specific. Similar behaviours shown by older dogs were described as ‘deliberate’. Both positive and negative experiences of dog ownership were identified. Free-text survey responses are a useful resource for exploring data but should be interpreted cautiously, as not all respondents answer these questions.

**Abstract:**

Owners’ understanding of dog behaviour influences dog welfare. This study aimed to investigate owners’ experiences of living with dogs and perceptions of dog behaviour/behaviour change. Data from an ongoing UK/ROI longitudinal study of dogs were used. Open-ended survey data (*n* = 3577 comments, *n* = 1808 dogs) when dogs were 12/16 weeks (data combined), 6, 12, 18 and 24 months were analysed to cover the dog’s puppyhood/adolescence. To evaluate the usefulness of open-ended survey questions, both quantitative textual and qualitative thematic analyses were employed. Textual analysis identified an overall positive sentiment at all timepoints; the proportion of positive: negative sentiments increased with the dog’s age. Words related to ‘love’ were the most frequent descriptors at all but the first timepoint, when ‘bite’ was the most frequent descriptor. Qualitative analysis helped to identify that owners attribute dog behaviour to ‘Dog’s biology’, ‘Personality/deliberate action’ and ‘External influences’. Analysis of open-ended survey responses helped to identify changes in perception over time. When dogs were young, owners described problematic behaviours as ‘mischievous’, unintentional and context-specific. Similar behaviours shown by older dogs were seen as ‘deliberate’. Both positive and negative experiences of dog ownership were identified. However, as not all respondents answered open-ended questions, the generalisability of our findings is limited.

## 1. Introduction

Owner perceptions and understanding of dog behaviour can influence the day-to-day management of dogs, potentially affecting dog welfare. For example, although repetitive behaviours, such as tail chasing, are often considered as indicative of poor welfare [1,2,3,4,5] or health issues [6], these behaviours are frequently described as funny or cute by dog owners and observers, potentially hindering help seeking by owners [7]. It is therefore important to understand how owners perceive dog behaviour. 

Owners’ perceptions of dog behaviour are particularly important when dogs are young, as dogs aged under two years old are most likely to be relinquished to shelters [8,9,10], and dogs under three years of age are most likely to be euthanised for behavioural reasons [11]. Although the exact reasons for a peak in relinquishments and euthanasia at this age are unknown, it is plausible that dog behaviour during adolescence contributes to a breakdown of the dog–owner bond. Dog behaviour continues to change across the dog’s life, but the degree of change is most pronounced during adolescence [12]. Adolescence is a relatively long period of development during which a juvenile becomes an adult and is marked by wide-ranging neurological and hormonal changes [12]. There is no precise agreed age at which an individual dog can be considered behaviourally mature, but cognitive and behavioural changes suggest that dogs between 6 months and 2 years of age can be considered as adolescent [13,14,15]. 

The adolescent period of development in mammals is typically associated with changes in social behaviour that include a weakened ability to regulate emotions and behaviour, resulting in increased impulsivity, reactivity to stressors, risk-taking behaviour and a greater awareness of conspecifics [12]. Owner–dog relationships have many features in common with human relationships and are believed to be based upon similar behavioural and hormonal bonding mechanisms [16,17]. In terms of the owner–dog relationship, dog behaviour during adolescence also bears perceptive similarities to that of human adolescent–parent relationships, with dogs displaying a socially specific reduction in obedience at this time for previously well-established cues such as ‘sit’ and an increase in separation-related behaviour problems [15]. Although anecdotal information exists, little scientific research has been undertaken on canine adolescence behaviour and owners’ experiences of dog adolescence. Therefore, the first objective of this study was to explore experiences of dog ownership as dogs mature between the age of 12 weeks and 2 years.

Previous research based on ‘Generation Pup’ longitudinal study data used here showed that both owner and dog characteristics influence whether a particular dog behaviour is perceived as problematic when dogs are 9 months of age [18]. Perception and interpretation of dog behaviour may also contribute to owner resilience in response to potentially problematic dog behaviour. To explore this further, a better understanding of owner perceptions of dog behaviour is needed. Therefore, the second objective of this study was to examine owners’ perceptions and attributions of dog behaviour as dogs mature between the age of 12 weeks and 2 years. To this end, the study utilises free-text responses from a longitudinal survey-based study, including responses to an ‘any other information’ question posed at the end of surveys at different timepoints. Both qualitative thematic analysis and quantitative textual analysis are used, and the third objective was to compare the two analytical approaches for the analysis of open-ended survey questions.

## 2. Materials and Methods

### 2.1. Study Design and Participants

This study used data collected as part of ‘Generation Pup’—a longitudinal study of dog health, behaviour and welfare. ‘Generation Pup’ is open to participants who are residents of the United Kingdom (UK) or the Republic of Ireland (ROI); aged 16 years or over; and who own a puppy of any breed or mix-breed. The study does not specify exclusion criteria related to the way dogs were acquired, as long as at the time of registration, the dog was younger than 16 weeks of age (or younger than 21 weeks if a puppy entered the UK/ROI through quarantine). Study recruitment is ongoing and will pause when 10,000 dogs are enrolled. Participants are recruited through social media, radio interviews, advertisements at veterinary practices, dog training venues and articles in veterinary, dog-related and other publications. This analysis uses data for dogs recruited between May 2016 and February 2020, i.e., before the COVID-19 pandemic. For further details regarding the methodology and protocol of ‘Generation Pup’, please see Murray et al. [19]. 

### 2.2. Data Collection

Data were obtained from online and postal self-administered surveys completed by owners when dogs were 12 and/or 16 weeks (depending on the age of the puppy when the owner joined the study), 6, 9, 12, 18 and 24 months (2 years). Owner and dog demographic characteristics were extracted from a survey completed upon registration for the project. Each survey has between 2–19 sections. The introductory surveys completed as a part of registration (1–3 weeks after acquisition or until 16 weeks of age) collect information about the owner, household, puppy and puppy acquisition. Topics covered in the later surveys analysed here include: introducing the puppy to the household, the puppy’s/dog’s experiences, activities undertaken with the puppy/dog, meeting other people, meeting other dogs, the puppy’s/dog’s behaviour, the puppy’s/dog’s day, the puppy’s sleep, diet, training approaches, health, surgery, neutering, insurance, the dog’s boarding/kennelling experience, breeding, exercise, mobility, reflection and other information. Topics are repeated at regular intervals; topics related to dog behaviour, the dog’s day, health and other information were included in all surveys analysed here (please see [19] for details). Each survey includes primarily close-ended questions and free-text boxes that enable owners to expand on and clarify or to provide an alternative response to the close-ended questions. Open-ended questions included in the analysis are specified in Table 1.

It has been suggested that open-ended questions within a survey can be used to explore general experiences and reflections related to other topics covered in a survey [20]. For this reason, the ‘any other information’ question was selected from all but the last timepoint of interest. This question was optional. Questions about the best, the funniest and the most annoying thing about one’s dog, asked only in the 2-year survey, were included, as they explicitly enquire about owners’ experiences and enable insight into owners’ perceptions of dog behaviour. A preliminary reading of responses to the ‘any other information’ question highlighted frequent comments related to the things owners enjoyed about their dogs. Therefore, a free-text question about behaviours that owners found to be a problem was included to learn more about behaviours owners find challenging and to explore perceptions of these behaviours. Owners were asked to answer this question if they specified that their dog shows a behaviour they find to be a problem. 

### 2.3. Data Analysis

Quantitative text analysis (word frequency, word importance and sentiment analyses) and qualitative thematic analyses were carried out. A mixed-methods approach was chosen to facilitate analytical triangulation [21]. Additionally, whereas the qualitative analysis lends itself to identifying owners’ perceptions and experiences, the quantitative textual analysis helps to quantify the changes in word use linked with these attributes over time. As open-ended survey questions were previously described as valuable but difficult to analyse [20], a comparison of the two analytical approaches to the analysis of this type of data is therefore valuable. Ahead of analyses, one dog from each multi-dog household was randomly included to avoid household-level clustering. Data from 12- and 16-week surveys were combined, as some owners joined the study in time to complete only the 16-week survey. 

#### 2.3.1. Quantitative Text Analysis: Data Preparation

To minimise the bias in sentiment introduced by asking specifically about problematic behaviours, only the text entered in response to ‘Any other information’ was analysed. This question was placed at the end of each survey and was unrelated to other questions about health, behaviour, the dog’s experiences, etc. Analysis of individual questions was further limited to the first five surveys (combined 12- and 16-week, 6-, 9-, 12- and 18-month surveys), as for the 2-year survey, we extracted questions specifically asking about owners’ experiences, which were not asked earlier. In addition, data from owners who answered questions at all timepoints (up to and including age 18 months) were included in a subset and analysed separately. The analysis for this subset of dogs was compared to the analysis based on the whole dataset to assess whether owners who completed all surveys described dog behaviour differently (i.e., with a different overall sentiment).

All textual analysis was carried out in R [22]. Data were prepared by removing punctuation, gaps and numbers and by transforming all words to lower case. Stop words, i.e., common English words that occur in any text frequently [23], were removed using pre-specified list words from the tm package [24]. Custom stop words associated with dogs (e.g., ‘dog’, ‘pup’, ‘bitch’, etc.) were also removed. Words were stemmed (i.e., reduced to their roots by removing suffixes and prefixes) using the SnowballC package [25] to improve word retrieval and recognition. For example, the words ‘connection’, ‘connections’, ‘connective’, ‘connected’ and ‘connecting’ were stemmed to ‘connect’. Stemming an algorithm works by conflating words with the same meaning rather than words that just have common linguistic roots (e.g., awe and awful have different stems). For this reason, stemming reduces some words in a non-obvious way, and some stems do not correspond to linguistic word stems, e.g., ‘happy’ and ‘happiness’ are stemmed as ‘happi’ to differentiate from stems of words related to ‘happening’ or ‘happened’ (which stem to ‘happen’).

#### 2.3.2. Quantitative Text Analysis: Word Frequency and Importance Analyses

To identify the most frequently used words at each timepoint (term frequency, TF), text was tokenised, i.e., converted to a format of one term per line. The term frequencies for tokens were calculated and depicted using word clouds with the Wordcloud package [26]. To evaluate the importance of a word with regard to a particular timepoint, the term frequency inverse document frequency (TF-IDF) metric was used. Inverse document frequency is derived by dividing the total number of documents (in this case, 5 documents corresponding to the first 5 timepoints) by the number of documents that a given word appears in and multiplying these two scores [27]. High scores reflect words that appear frequently in a few documents and are therefore important to those documents, and low scores identify words that appear frequently in every document and, as such, are less important [27].

#### 2.3.3. Quantitative Text Analysis: Sentiment Analysis

Sentiment analysis was carried out using multiple lexicon dictionaries (NRC, Bing, AFFIN, Syuazhet) to reduce uncertainty and error related to relying on one lexicon. The NRC Emotion Lexicon categorises English words into eight basic emotions (anger, fear, anticipation, trust, surprise, joy, sadness and disgust) and two binary sentiments (positive and negative) [23]. The Bing [28] and Syuazhet [29] lexicons categorise words as having a positive or negative sentiment (i.e., as binaries), and the AFFIN lexicon does so by assigning a score from −5 (most negative) to 5 (most positive) [30]. Sentiment scores from individual tokens were added for each survey response, and to compare different lexicon dictionaries, the net sentiment (positive–negative) was plotted. The proportion of positive to negative sentiments was also ascertained using the Syuzhet lexicon. Words that contributed the most to the overall positive and negative sentiments at each timepoint were identified.

#### 2.3.4. Qualitative Analysis: Data Coding and Thematic Analysis

Thematic analysis was applied to longitudinal data [31]. The purpose of thematic analysis is to identify the patterns within a text, i.e., themes [32]. Thematic analysis is a flexible approach that is suitable for characterising a range and diversity of perceptions, beliefs, experiences, representations, etc. but not for the quantification of findings [33]. Thematic analysis is also not designed to establish ‘the truth’. Consequently, as the respondents were free to discuss any behaviours as problematic to them, the findings capture owner perceptions of dog behaviour, which should be distinguished from categories of behaviours based on clinical or ethological research. 

After familiarisation with the text, all questions listed in Table 1 were coded inductively line-by-line by one researcher (S.C.O.-G). Additionally, 15% of the text was coded by two co-authors (K.G.) to ensure coding rigour. Inductive coding means that codes were based on the data rather than being developed before the analysis. At the same time, coding was focused on identifying experiences of dog ownership and perceptions and attributions of dog behaviour. Codes were refined and revised as coding progressed and in the course of discussion between the co-authors [34]. Themes were first identified for individual timepoints, and then comparisons were made between timepoints [31]. Themes were constructed by grouping and categorising codes and then defining the code groups. Next, between-timepoint comparisons were made. This was achieved by identifying differences and similarities between timepoints with respect to owners’ attributions of changes in dog behaviour and the prominence of owners’ experiences [31,35,36]. Comments regarding changes in dog behaviour over time are provided alongside themes and illustrative quotes. All coding was conducted in NVivo 11 software [37].

### 2.4. Research Ethics Statement

The study had ethical approval from the University of Bristol Animal Welfare Ethical Research Board (UIN/18/052), the Clinical Research Ethical Review Board at the Royal Veterinary College–URN 2017 1658-3, the Social Science Ethical Review Board at the Royal Veterinary College–URN SR2017-1116, and Dogs Trust Ethical Review Board–ERB009. Informed consent to take part in the study was obtained from all participants. During the analysis and when selecting illustrative quotes, data were pseudonymised by removing identifiable characteristics (such as the dog’s or owner’s names or the mention of breeds, if they were uncommon).

## 3. Results

### 3.1. Sample Description

After removing (at random) all dogs bar one from multi-dog households, 3577 comments about 1808 dogs were included in the analysis. Most (88.9%, *n* = 1609) owners identified as female, 10.8% (*n* = 195) identified as male, and this information was unavailable for 0.2% (*n* = 4) of respondents. The most common age of respondents was 35–44 years and 45–54 years (21.9%, *n* = 396 and 22.3%, *n* = 403 respectively). There was an even split between male (*n* = 907, 50.2%) and female dogs (*n* = 901, 49.8%). Slightly over half the dogs were crossbreeds or mongrels (*n* = 978, 54.0%), and 46.0% (*n* = 830) were purebred (for further details about the cohort, see [19]). No obvious differences in comments were identified between the full dataset and the subset of 117 dogs for whom data at all timepoints were available. No differences with respect to demographic data were identified between all participants registered onto the study at the time of data analysis (*n* = 3162) and participants who answered questions analysed here.

### 3.2. Word Frequency

Aside from the 12–16-week and the 9-month mark, at all timepoints, words related to ‘love’ were most frequent (Figure 1). At 12–16 weeks, the most common words were ‘bark’, ‘bite’, ‘jump’ and ‘play’. At 9 months, words related to ’love’ were relatively frequent, but the most frequently used words were common words, such as ‘get’ or ‘week’ (Figure 1).

### 3.3. Word Importance: TF-IDF

The TF-IDF analysis (Figure 2) shows that the most important words at 12–16 weeks were related to puppy biting/mouthing and chewing (‘bite’, ‘ankle’, ‘nip’, ‘furniture’, ‘trouser’, ‘cloth’- stem of clothing, ‘mouth’- stem of mouthing), vaccinations and microchipping. At 6 months, important words were related to dog behaviour (‘teeth’- stem of teething, ‘bite’, ‘chew’), adolescence (‘adolescence’, ‘test’- stem of testing), dog temperament/disposition (‘calm’, ‘pleasure’, ‘eager’) and owners’ expectations (‘anticipat’- stem of anticipate, ‘regret’, ‘problematic’). Important words at 9 months reflected possible changes within the family (‘university’, ‘students’), the dog’s health (‘itch’- stem of itchiness, ‘hypoallergen’- stem of hypoallergenic, ‘feed’- stem of feeding) and dog behaviour (‘unabl’- stem of unable, ‘unsettl’- stem of unsettled). At 12 months, the most important words were related to the dog’s age (‘birthdai’- stem of birthday), dog behaviour (e.g., ‘destroy’, ‘immatur’- stem of immature, ‘began’) and breed (‘shepherd’, ‘mini’, ‘mali’- stem of Malinois breed). The variation in responses at the age of 18 months was lower than at other timepoints, possibly due to a smaller sample size (*n* = 138). At 18 months, the most important words were also associated with dog breed (‘lhasa’—related to Lhasa Apso), dogs’ activities, their character and behaviour (‘squeaky’, e.g., playing with a squeaky toy, ‘skittish’, ‘humour’, ‘capac’- stem of capacity, ‘emot’- stem of emotion, ‘capabl’- stem of capable) and, plausibly, things that owners and dogs do together/encounter on walks and holidays (‘trailer’, ‘pheasant’, ‘lure’).

### 3.4. Sentiment Analysis

All lexicons apart from Bing (at all timepoints) and Syuzhet (at 12–16 weeks) showed a net positive sentiment (Figure 3a). The trend across all lexicons shows that the proportion of positive to negative sentiment was the lowest in the 12–16-week survey, and that the sentiment was more positive with each consecutive timepoint (Figure 3).

Words that contributed most to the negative sentiments were related to dog behaviour (e.g., ‘bite’, ‘whine’, ‘naughty’, ‘unsettl’- stem of unsettled), changes in dog behaviour/the owner’s expectations (e.g., ‘slow’, ‘hard’, ‘unable’, ‘difficult’, ‘concern’) and health (e.g., ‘sick’, ‘lame’, ‘wound’, ‘pain’). Words that made the most significant contribution to the positive sentiment were related to dogs’ dispositions (e.g., ‘happi’- stem of happiness, ‘love’- stem of lovely, loving, loved and love, ‘calm’, friend’- stem of friendly) and behaviour change/training (e.g., ‘good’, ‘better’, ‘great’, ‘work’, Figure 4).

### 3.5. Qualitative Analysis

Three themes were constructed: ‘Explaining dog behaviour’, ‘Positive experiences of dog ownership’ and ‘Negative experiences of dog ownership’.

#### 3.5.1. Explaining Dog Behaviour

Some explanations of behaviour did not change over time, e.g., references to the dog’s biology. At all timepoints, some owners explained dog behaviour with respect to the dog’s age; however, the references changed over time (from ‘toddler’ in the 12–16-week survey to a ‘teenager’ later on). In the 12–16-week, 6-, 9-, 12- and 18-month surveys, owners often saw behaviours as transient—something a dog will ‘grow out of’. In the 12-month, 18-month and 2-year surveys, owners tended to describe dog behaviour as ‘immature’ or expressed their frustration when a dog was ‘still’ engaging in a particular behaviour. In the 6-, 9- and 12-month surveys, explaining dog behaviour with respect to hormones was more common than at other timepoints. Explanations that attributed behaviour to the dog’s personality or to deliberate action by the dog were more common in the later surveys (those at 12 and 18 months and 2 years, see Table 2 for details).

#### 3.5.2. Positive Experiences of Dog Ownership

The shift in the theme of ‘Positive experiences of dog ownership’ over time was subtle, and similar sub-themes and codes were identified at all timepoints (see Table 3).

#### 3.5.3. Challenges of Dog Ownership

Most sub-themes within the ‘Challenges of dog ownership’ theme were identified at all timepoints (Table 4). However, the negative experiences mentioned in earlier surveys (12–16 weeks) differed from those discussed later. Four sub-themes were identified: ‘Settling into a life with a family’, ‘Issues with training’, ‘Challenging personality/disposition’ and ‘Challenging behaviour’ (see Table 4).

## 4. Discussion

### 4.1. Owners’ Attributions of Dog Behaviour

Three main patterns in attributions of dog behaviour were identified: (1) referring to the dog’s biology, such as breed, genetics, sex and age; (2) seeing dog behaviour as reflective of the dog’s personality/deliberate action; and (3) perceiving behaviour as externally influenced. Breed-related explanations of dog behaviour were common, and references to rare breeds were identified as important words through TF-IDF analysis, particularly in later surveys (12 and 18 months). This shows that though textual analysis is helpful at identifying important words in a document, context (in this case, the findings of the thematic analysis) may be needed for interpretation. Both traits that owners enjoyed (such as sociability and friendliness) and those that owners found challenging, such as nervousness, were seen as breed-related. Explanations that referred to the dog’s biology (e.g., genetics) were sometimes used to explain a lack of expectation that the dog’s behaviour would improve and a cessation of attempts to change the dog’s behaviour. The literature on parent–child relationships shows that parents who attribute their child’s behaviour to stable, internal causes (genetics, free will) feel helpless and believe that their child’s behaviour is unlikely to improve [38,39]. Further research is needed to explore whether the same pattern holds true for human–dog relationships.

A number of behaviours, in particular those experienced as challenging, were seen as deliberate and a reflection of the dog’s personality. For example, owners described things that dogs did despite previous training or after a period of improvement e.g., pulling on the lead, not coming back when called or toileting within the home. Research into parent–child relationships shows that attributing responsibility for the behaviour to a child’s will is related to greater anger and emotional arousal in parents compared to the same behaviour being explained with situational attributions [38,40]. It is unclear whether the same is true for human–dog relationships, and this theme warrants further studies.

Many dog behaviours were explained with external factors, such as the impact of other dogs, people, past training or socialisation. In particular, potentially serious behaviour problems, such as dog-to-dog reactivity or aggressive behaviour towards people or other dogs, were discussed in this way. Existing research shows that attributing children’s misbehaviour to situations outside of the child’s control helps caregivers to maintain a positive view of the child [39]. This style of attribution is also associated with reduced conflict between parents and children [39]. Again, it is necessary to establish whether the same pattern applies to the dog–human relationship. Nonetheless, as previous studies show that owners often experience stigma and feel blamed for their dog’s behaviour [41,42], contextual explanations may help owners to cope and encourage them to seek help if needed.

At all timepoints, owners’ comments indicated a misunderstanding of the dog’s behaviour. For example, some owners explained that a dog was not aggressive, because they were wagging their tail. Research shows that tail wagging alone is a poor indicator of a dog’s emotional state, as it is more likely to indicate an overall arousal [43]. A few owners discussed spinning behaviour as something funny, despite evidence indicating that repetitive behaviour may be associated with poor welfare [1,2,4,5]. Some possibly dangerous behaviours, such as the puppy chewing and swallowing stones or socks, were described as normal and seen as‘mischievious’. In these cases, further education into dog behaviour is thus needed.

Finally, the use of anthropomorphic language (e.g., calling dogs ‘wimpy’ or ‘needy’ when discussing difficulties with leaving them on their own, ‘grumpy’ when talking about reactivity, describing disobedience as ‘testing boundaries’, referring to dogs as a ‘toddler’ or ‘teenager’ or talking about their ‘mischief’, ‘sass’ and ‘stubbornness’) was common. Anthropomorphic references could reflect a lack of language to describe dog behaviour or owners’ perceptions that dogs share human-like inner lives and motivations. The latter explanation could indicate that owners assume that dogs’ motivations for challenging behaviours are similar to humans. This could lead to unrealistic expectations regarding dog behaviour. However, anthropomorphic language also encourages empathy towards animals [44], which could help owners in seeking help for their dog’s behaviour if needed. Together, these findings suggest that further research is needed to ascertain a relationship between the pattern of attributing dog behaviour and owner–dog interactions.

### 4.2. Change in Dog Behaviour

Textual and qualitative analyses show that at all timepoints, some owners were concerned about their dog’s training, but they also derived a lot of satisfaction from it. At 12–16 weeks, owners often remarked how quickly their puppies became housetrained and settled into family life, and textual analysis identified the word ‘train’ among the most often used words at this timepoint (Figure 1). At the same timepoint, owners also discussed issues with housetraining and their dog’s ability to adapt to family life. Words related to biting, mouthing and nipping were identified as important in the 12–16-week surveys but not at other timepoints (except for a single word, ‘bite’, a stem of biting, which was identified as important in the 6-month survey). This suggests that the challenges related to living with puppies are different from the challenges related to living with older dogs. Issues related to training, puppy biting and mouthing and settling in at night were the likely reason that the overall sentiment in the 12–16-week surveys was the lowest.

Issues with housetraining were discussed in later surveys as well. However, whereas at the earlier timepoints, owners emphasised that a puppy was learning (i.e., the behaviour was explained as transient and modifiable), later (from 6 months onwards), they stressed that the dog was still engaging in the inappropriate behaviour (i.e., lesser emphasis on the behaviour being temporal and requiring training to change). In earlier surveys, jumping up or pulling on the lead were more often described as a ‘work in progress’. Later, owners emphasised that their dog continued to engage in these behaviours despite previous training, or that the behaviour deteriorated after a period of improvement, and the behaviours were more often seen as a part of the dog’s personality. This suggests that owners may accept undesirable behaviours when they are perceived as specific to a dog’s age but find the same behaviours more challenging when the dog is older. Comments regarding the hope that a dog will ‘grow out’ of these behaviours indicate that some owners expect their dogs to toilet appropriately, walk on a loose lead and/or be less excitable by a certain age and may not be aware that further training in those areas is required to change behaviour. Further communication of known changes in dog behaviour around adolescence [15] is needed to manage owner expectations.

At 6 months, words associated with behavioural changes relating to adolescence (‘adolecen’- stem of adolescence, ‘test’- stem of testing) emerged as important in showing how central these behaviours were to owners’ experiences at this point. In the 6-month, and to a lesser degree, in the 9-month surveys, dog owners still struggled with puppy biting and mouthing but also reported that their dogs were barking more than previously and were more ‘mischievous’ and disobedient, which echoes previous research into adolescence in dogs [15].

From 6 months onwards, the words ‘aggression’ (or related words such as ‘growl’) were identified as contributing to the negative sentiment, and aggressive and/or reactive behaviour emerged as a concern through qualitative analysis. Overall, comments about the dog’s aggression and/or reactivity towards known and unfamiliar people and guarding were, however, uncommon. Research shows that the risk of aggression towards familiar and unfamiliar people increases with a dog’s age [45] and is more common in adult dogs than in adolescent or senior dogs [46]. Aggressive behaviours are likely to emerge later as dogs have more negative experiences with people and escalate their responses over time. Descriptions of intraspecific aggression and/or reactivity also became more common from 12 months onwards. Previous research has identified that the risk of intraspecific aggression increases with age, possibly as a result of dogs accumulating more negative experiences around other dogs [47]. In addition, behavioural changes related to dogs reaching social maturity around the 12-month timepoint [12] may impact upon their interactions with other dogs, contributing to the observed findings.

Over time, a gradual increase in positive sentiment was identified. Comments about dogs being affectionate and relaxed were more common from 12 months onwards. It is possible that calmer behaviour, lesser intensity of behaviours associated with adolescence and the development of a bond with the dog explain the more positive sentiment over time. In the 12–16-week survey, owners commented on how quickly dogs adapted to family life. In the later surveys, doing things together (e.g., walking, playing, going to the pub) was more often discussed as an important part of the owner–dog relationship. In the 12–16-week and 6-month surveys, owners more often commented on their dog’s cuteness. Previous research shows that owner-perceived dog cuteness is a significant factor that is predictive of the quality of the human–dog relationship [48]. Factors that contribute to bonding with a dog have previously been identified [49], but little is known about the initial development of this relationship and how the dog–owner relationship changes over time.

Some behaviours and dog characteristics observed at all timepoints were seen as both positive and negative aspects of dog ownership. For example, some owners enjoyed their dog’s ‘mischief’, ‘sassiness’, ‘enthusiasm’ and ‘affection’ reflected in behaviours such as ‘stealing’ and destroying some household items, being excitable and seeking the owner’s proximity. These behaviours were often framed as signs of a dog’s unique character and personality. However, owners complained about dogs ‘testing boundaries’, excessive excitability and difficulties leaving their dogs alone. It is possible that acceptance of dog behaviours depends, to an extent, on their context, intensity, the way the behaviour is framed and owner characteristics. For instance, jumping up at people as well as ‘stealing’ household items may be seen as ‘mischievous’ and ‘cute’ when a dog is young, but these behaviours become a nuisance as the dog matures or gets hold of valuable items. Alternatively, this finding may suggest that over time, perception of the same behaviour changes, and behaviours that were initially tolerated are perceived as not acceptable later on. Different levels of understanding the reasons for dog behaviour can also explain why similar behaviours were discussed as both positive and negative. For example, when explaining why dog barking is irritating, owners emphasised that their dogs barked ‘for no reason’; not understanding the reasons for this behaviour contributes to seeing it as annoying. Positive framing of undesirable behaviours was previously observed as a helpful coping strategy [41]. Here, labelling challenging behaviours as mischievous may be a way of humouring the dog helping to accept the dog’s behaviour.

### 4.3. Positive Experiences Related to Dog Ownership

This study identified an overall positive sentiment in comments about dogs at all timepoints. Across all timepoints, words derived from ‘love’ were among the most often used terms (Figure 1 and Figure 4), and the relationship with a dog was an important sub-theme identified through qualitative analysis. The relationship with the dog was often described as life-changing, and owners expressed that their dog helped to reduce their loneliness and gave them a purpose in life. This echoes past studies that identified the impact on interpersonal relations, happiness and the owner’s wellbeing as important facets of the owner–dog relationship [50]. The textual analysis showed that words related to a dog’s friendly, happy, sociable and calm disposition as well as training progress made the greatest contribution to positive sentiment. The happiness, enthusiasm, affection, zest for life and trainability shown by dogs were also important sub-themes identified through qualitative analysis, showing that the findings from the textual and thematic analyses were similar. Compared to the previous research into characteristics of an ideal dog in Australia [51], we did not find owners wanting their dogs to be faithful or protective. Our findings were, however, in line with the description of an ideal dog in Italy [52], where being calm, sociable, healthy, well trained, adaptable, energetic and easy to manage emerged as ideal behavioural traits. The differences between these and previous findings could be explained with differences in study methodology—we did not explicitly ask about the characteristics of an ideal dog; this analysis was based on ‘other information’ comments at the end of the survey.

### 4.4. Negative Experiences Related to Dog Ownership

At all timepoints, some owners described difficulties related to leaving their dog alone, as well as to their dog’s attention-seeking behaviour and jealousy (which described times when dogs reacted negatively to a lack of attention). No changes over time were identified for these sub-themes. This contrasts with previous research in which separation-related behaviours were found to worsen around adolescence compared to the pre- and post-adolescent periods [15]. This difference in findings could be explained by the fact that the previous study used specific measures of attachment to the carer, whereas in the current study, owners described difficulties related to leaving the dog alone.

At all timepoints, dog owners found a dog’s ‘stubbornness’ difficult. The non-academic literature frequently portrays some breeds as inherently stubborn [53]. Some dog personality traits have a strong genetic component and could therefore be expected to be observed early on and remain relatively stable during the dog’s life [54,55]. However, behavioural responses are likely to result from intersectional effects of multiple personality characteristics as well as learned experience. It is also plausible that a consistency of this aspect as an issue across all timepoints reflects the owners’ struggle to identify how to motivate their dogs to behave in the ways they desire.

### 4.5. Strengths and Limitations

Open-ended survey data and ‘any other information’-type questions in particular are valuable, but these are a hard-to-analyse and under-utilised resource [20]. Methodological guidelines for analysing general, non-directive open-ended survey questions are scant [20,56]. The application of a mixed-method approach (i.e., using qualitative thematic and quantitative textual analyses) helps to develop analytical triangulation [21], allows for cross-checking the findings and makes the presented results more robust. To the best of our knowledge, this is also the first analysis of dog owner reported experiences and attributions based on longitudinal data. The data presented here are unlikely to be affected by a recall bias, as surveys were administered at regular intervals asking for recent experiences.

However, our findings have a number of limitations. The analysed text did not specifically ask about owners’ experiences and perceptions of dog behaviour (except for the questions from the 2-year survey, which explicitly asked about the best, most annoying and funniest aspects of dog ownership). As such, our data were not always rich enough to understand the details of owners’ experiences. Future research could draw on in-depth qualitative interview data repeated regularly over the course of a dog’s life to develop more detailed understanding. This approach could help to explore the nuances of change in the human–dog relationship experienced as dogs mature. Moreover, as the questions analysed here were placed at the end of the survey, it is possible that the analysed text was influenced by the content of previous questions. For example, the 9-month survey asked a number of questions about a dog’s surgeries, and it is possible that health-related concerns identified through textual analysis for this point were a result of this bias.

Although textual analysis is quick to carry out, it is hard to interpret its findings out of context. For example, the word frequency analysis identified that words related to ‘love’ were common at all time points, which was further corroborated by the qualitative analysis (sub-theme of ‘Dogs love and bond with a dog’). However, other words were also common, e.g., ‘get’ and ‘week’ (Figure 1, 9-month survey). This suggests that though word frequency graphs may be useful for highlighting macro-trends in word use over time, they need to be interpreted cautiously. Moreover, as the algorithm reduces words to stems to enable comparison, without context, it is unclear whether the high frequency of the stem ‘love’ is due to reporting dogs as loved, talking about love for a dog or describing them as lovely. Some of the most important or frequent words identified through the textual analysis could be used in a negation. Combining the findings of qualitative and textual analyses helps to reduce this risk. In addition, TF-IDF analysis of the 18-month timepoint showed less variation in the importance of words than analysis of previous timepoints, due to a lower sample size at 18 months compared to previous timepoints (*n* = 138), indicating how this approach is limited by sample size. Finally, the sentiment lexicons are also trained on data different from that analysed here (e.g., data from film reviews) and may therefore not capture the sentiment in our text accurately [57]. We used multiple lexicons (trained on different datasets) to overcome this issue; however, our findings need to be interpreted cautiously.

The textual analysis of the sub-set of comments for dogs for whom data at all timepoints were available did not reveal different patterns than the analysis of comments for the whole dataset. However, the sample size of this subset was small. The ‘any other comments’ question was not compulsory, and the question about problematic behaviour was asked if a problem was first reported. It is possible that only the owners who felt particularly strongly about their experiences offered comments, biasing the findings. The ‘Generation Pup’ cohort may also not be entirely representative of the UK dog-owning population. Surveys require a substantial time commitment, so it is possible that our findings reflect the experiences and perceptions of particularly dedicated dog owners.

## 5. Conclusions

This exploratory analysis showed that experiences related to dog ownership over the first two years of a dog’s life are primarily positive, as reflected in the overall sentiment of owners’ comments. Owners enjoyed forming a relationship with their dog and found their dog’s personality, training and physical appearance rewarding. Over time, positive experiences of dog ownership changed less than negative experiences, which were mostly related to a dog’s age. Many owners found early experiences with puppies and behaviours associated with adolescence challenging. This finding can be used to manage expectations of new or prospective dog owners. Owners sometimes believed that adolescence-related behaviours would resolve themselves without an intervention as dogs matured. Although this perception can help the owner to tolerate challenging behaviour, it can also hinder their seeking help. Therefore, further owner education regarding age-specific changes in dog behaviour and training needs may be beneficial. The most common pattern of dog behaviour attribution referred to a dog’s unmodifiable characteristics, such breed, genetics or the dog’s personality. Other attributions focused on external or situational factors, such as the dog’s training or the social influence of other people and dogs. Most behavioural explanations were similar at all timepoints; however, a shift in the attribution of dog behaviour to their personality was observed as dogs matured. Attributions that explain dog behaviour with regard to internal characteristics can help owners to justify and accept undesirable behaviours, preserving the owner–dog bond. However, this perception can also discourage attempts to change these behaviours, potentially threatening the dog’s welfare. Many behaviours were perceived as both positive and challenging, likely depending on the context in which they were expressed, as well as their intensity, the dog’s age, the owners’ previous experience and their personality. Further exploration of the factors that shape owners’ attributions of dog behaviour and the ways in which these attributions may translate into the ways in which dogs are managed is needed. Finally, the study identified that the use of qualitative thematic and quantitative textual analyses to study free-text survey responses can be a useful way of exploring perceptions and experiences over time and a way to generate research questions. However, due to the limitations of this type of data, findings need to be interpreted cautiously.

## Figures and Tables

**Figure 1 animals-13-01863-f001:**
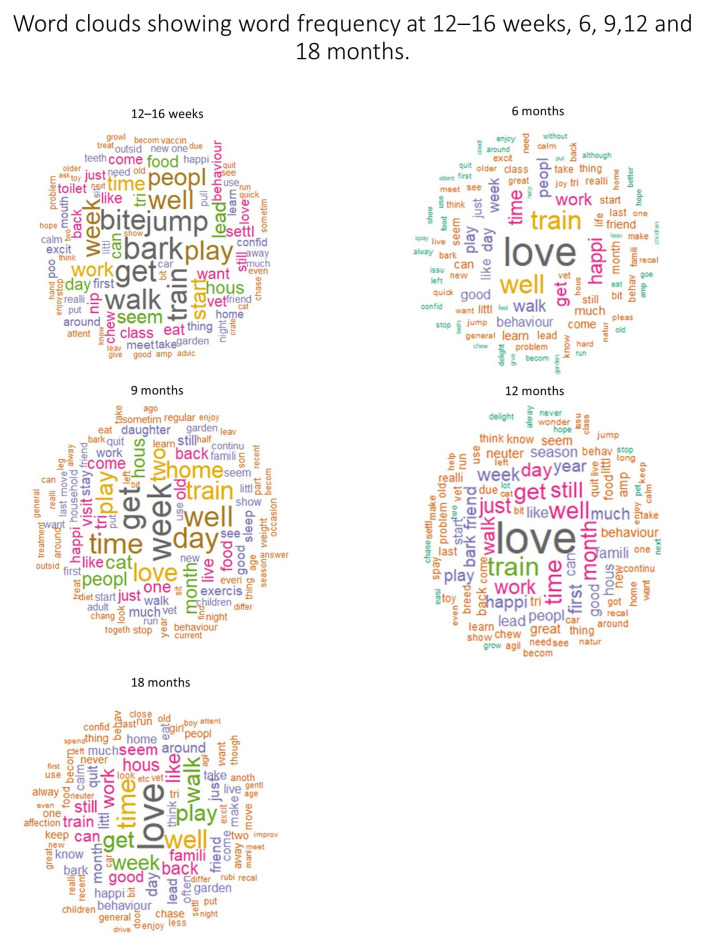
Word clouds showing the most frequently used words in responses to the ‘Any other information’ questions at 12–16 weeks, 6, 9, 12 and 18 months. Word size reflects frequency of use.

**Figure 2 animals-13-01863-f002:**
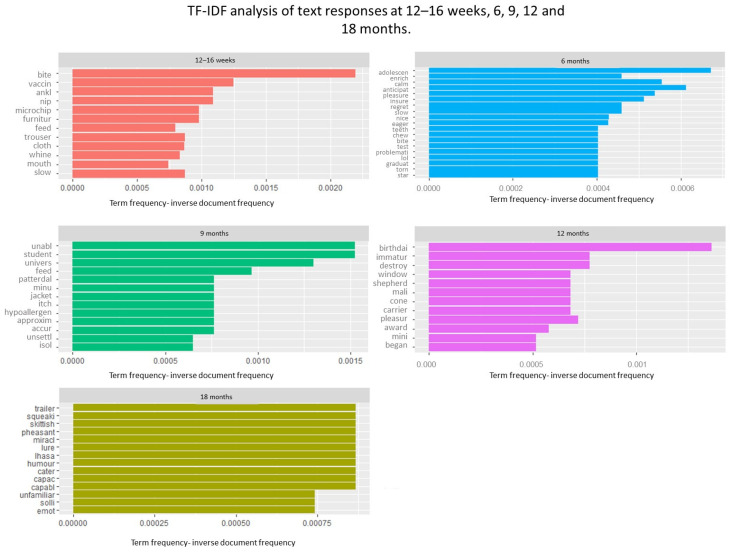
Word importance left in comments to ‘Any other information’ questions at 12–16 weeks, 6, 9, 12 and 18 months expressed using term frequency inverse document frequency (TF-IDF) metric.

**Figure 3 animals-13-01863-f003:**
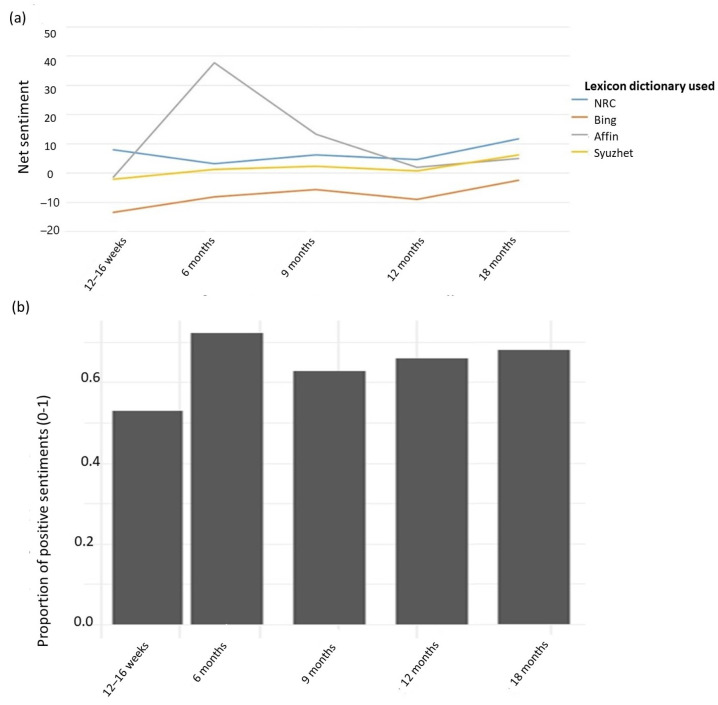
(**a**). Net sentiment (positive–negative) in words left in comments to ‘Any other information’ questions at 12–16 weeks, 6, 9, 12 and 18 months and (**b**). the proportion of positive sentiments in the whole text (0–1) according to Syuzhet lexicon.

**Figure 4 animals-13-01863-f004:**
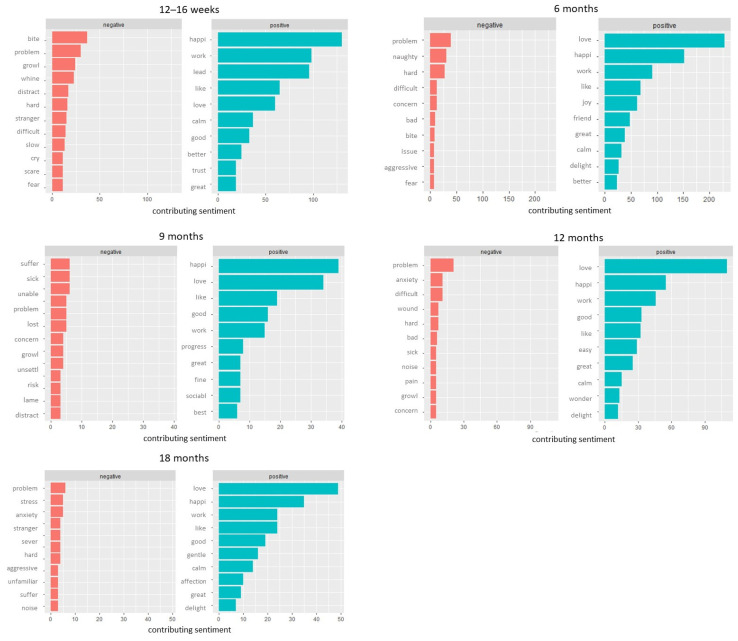
Word contribution to positive and negative sentiments left in comments to ‘Any other information’ questions at 12–16 weeks, 6, 9,12 and 18 months.

**Table 1 animals-13-01863-t001:** Timepoints, questions, number of responses available for analysis and the number of completed surveys at a given timepoint. All data were analysed with qualitative thematic analysis; data analysed with quantitative textual analysis are marked with *.

Timepoint	Survey Availability and Dog’s Age	Questions Included in the Analysis	Number of Complete Responses to the Question/Sample Size at a Given Timepoint (% of Respondents Who Answered the Question)
12 and 16 weeks (combined)	12 weeks: 84–108 days	Please describe the behaviour(s) that you find to be a problem.	1154/4427 (21.4)
	16 weeks: 112–136 days	Any other information? *	998/4427 (22.5)
6 months	180–204 days	Please describe the behaviour(s) that you find to be a problem.	494/1788 (27.6)
		Please use the space below to add any other information about your puppy that you would like to share with us. *	608/1788 (34.0)
9 months	274–316 days	Please describe the behaviour(s) that you find to be a problem.	467/1259 (37.1)
		Please use the space below to add any other information about your puppy that you would like to share with us. *	235/1259 (18.7)
12 months	365–407 days	Please describe the behaviour(s) that you find to be a problem.	463/1320 (60.6)
		Please use the space below to add any other information about your dog that you would like to share with us. *	337/1320 (35.1)
18 months	547–589 days	Please describe the behaviour(s) that you find to be a problem.	199/747 (26.6)
		Please use the space below to add any other information about your dog that you would like to share with us. *	138/747 (18.5)
2 years	730–772 days	The best thing about my dog is…	297/302 (98.3)
		The most annoying thing about my dog is…	274/302 (90.7)
		The funniest thing about my dog is…	271/302 (89.7)

**Table 2 animals-13-01863-t002:** ‘Explaining dog behaviour’ theme, sub-themes and codes. Quotes that illustrate a particular code are provided.

Code	Illustrative Quote
1. Dog’s biology	
a. Breed	*“She is already showing characteristics of her breed such as walking to heel, sniffing out in hedgerows, following us around the house.”* (12–16 weeks) *“She is a typical Patterdale in that she is constantly digging holes and hunting out mice around the garden”.* (9 months) *“[Dog’s name] is a very excitable dog but appears quite nervous. Anxiety is a recognised issue with Vizslas.”* (18 months) *“Her groomer said to me, ‘you do realise that schnauzers notoriously don’t “grow up” until they’re about 3 years old don’t you!’”* (2 years)
b. Genetics	*“[Dog’s name] had settled in really well and is noted by many people as a calmer Springer. How much of this is due to genetics- her mum is an assistance dog and very calm; how much is due to nurture- we are a quiet adult only house (children have left home) ...who knows?!”* (6 months) *“[Clinical behaviourist] told us that it is [dog’s name], not us that is behind her anxious, aggressive behaviour. We have done everything right- tried to socialise and habituate, trying to only have positive experiences for her, took her out and about, gentle introductions* etc. *but she ‘resisted’ all of this- largely genetics of the breed, her mother, and early experiences.”* (18 months) *“Unfortunately, although she is keen to work and very stylish her breeding has made her very wide and off-contact on sheep and thesis* [sic?] *almost impossible to correct since dogs get wider as they get older. It is frustrating because this is a genetic fault, not a training fault.”* (2 years)
c. Sex/hormones	*“Very solid temperament to date but bitches can change so we will see”.* (12–16 weeks) *“Has had a 3 month first season followed by a pseudo pregnancy so has been very hormonal, excitable and clinging plus has shown aggression to our other bitch”.* (9 months) *“He is also peeing around the house, occasionally on the bed. (…) I think he is marking… (…) Hopefully the neutering calms these behaviours”. (9 months)* *“Since her first proper season she has calmed a lot and is behaving so much better.”* (12 months)
d. Age	*“[Dog’s name] likes to bite and nip and lunge at your face when biting. I have been told this is normal puppy behaviour especially when teething. (…) [Dog’s name] is acting like a typical toddler”.* (12–16 weeks) *“Best way I can describe this is that he’s being sassy! Just normal ‘teenager’ behaviour as far as I am aware. Not following basic commands he definitely knows likes sit—recall has worsened—tearing up his bed—humping”.* (6 months) *“We are training lightly, but although he is a big dog, he is immature so we only do short spells of training.”* (12 months) *“Much more mature in last 6 weeks, no more digging in garden, rolling on dead things, eating nasty items, also much less worried about unfamiliar looking people”.* (18 months)
2. Dog’s personality/deliberate action	
	*“She’s definitely learnt what’s acceptable and what’s not although she’ll sometimes do it anyway but clearly knows she shouldn’t.”* (12–16 weeks) *“She had started in the last few weeks pushing the boundaries and ‘barking’ back when I tell her no or go to move her.”* (6 months) *“Pulls on lead despite loose lead walk training. Doesn’t ask to go out to toilet, relies on us to open door, and will have ‘accidents’”.* (12 months) *“He is very independent and will ignore recall completely most of the time”.* (18 months)
3. External influence	
a. Training and socialisation	*“We are desperately trying to stop the behaviour and when we say ‘off’ he does stop and we treat him. However he has now learnt if he bites and told to ‘off’ he gets a treat. Tricky—work in progress. Been told he’ll grow out of by a dog trainer.”* (12–16 weeks) *“After just a few weeks [of training/socialisation classes], I have noticed a big change in [dog’s name] attitude and confidence both in class and when out during walks.”* (6 months) *“Considering her breed specifics I’m really pleased with how [Dog’s name] is doing, I have worked hard at socialising her & teaching her tricks & games”.* (9- months) *“Her training is paying off and she is a lovely and generally well behaved dog (…)”.* (18 months).
b. Influenced by people	*“[M]ost of the time people encourage it by stroking him which annoys me and that’s why he keeps thinking its ok. At home I discourage it and he’s starting to listen but on walks strangers re-enforce it.”* (12–16 weeks) *“[Dog’s name] is shy and timid with other people. This goes back to 2 incidents which happened during his fear period at about 8—10 weeks, when he was picked up from the floor without warning (…). It happened so quickly that I couldn’t stop it and [dog’s name] really took offense.”* (6 months) *“When he doesn’t want to head back home after a walk he will grumble and try and nibble your feet. (He mainly does this to my husband as he is a soft touch)”.* (2 years)
c. Influenced by other dogs	*“Copying the poor behaviour in my other dogs—jumping up at the front gate when the postman comes”.* (12–16 weeks) *“Not sure if it’s because we have an older dog to help, but having [Dog’s name] has been far easier than our last puppy experience, she’s keen to learn and for a pup behaves very well”.* (6 months) *“[Dog’s name] is a confident dog—maybe because he has had the older dog as a role model and company.”* (18 months)

**Table 3 animals-13-01863-t003:** Sub-themes and codes that contribute to the ‘Positive experience of dog ownership’ theme at different timepoints. Quotes that illustrate a particular sub-theme/code are provided in addition to comments regarding the direction of change.

Code	12–16 Weeks	6 and 9 Months	12 and 18 Months	2 Years	Change Over Time
1. Relationship with a dog					
a. Dog’s love and bond with a dog	*“He is a lovely funny boy, and gives me hours of happiness”*	*“He has been and is a godsend. Would be totally lost without him even after only 4-5 months with us. He has given me a purpose and focus to each day.”* (6 months)	*“He is a lovely dog. Such a great nature and great with the children. He is very soppy!”* (18 months)	*“She fits into our family and helps us get out and about with the kids. Even on days we don’t want to leave the house! She has grown into such an important part of our family and has taught the children a lot in the short space of time we have had her. She is a wonderful dog and we all love her loads!”*	No change
b. Entertainment and fun	*“He’s a funny, little fellow, who makes me laugh”*	*“Makes us laugh too”* (6 months)	*“He keeps us amused and entertained”* (12 months)	*“There are certain things about dogs that make you laugh and she is hilarious (…) she loves stones to be thrown for her she* *goes mental and spins around!”*	No change
2. Dog’s personality/disposition					
a. Happiness	*“Very happy contented puppy”*	*“[Dog’s name]” is a very happy”* (6 months)	*“[Dog’s name] continues to be a happy friendly dog.”* (12 months)	*“(…) she grunts in pleasure when she is stroked* *(…) she literally jumps for joy on the beach, dancing with happiness”*	No change
b. Enthusiasm	*“He’s very (…) enthusiastic towards everyone and other dogs”*	*“She loves everything and has enthusiasm for life.”* (9- months)	*“She charges around like a ballistic missile much of time and is mega- enthusiastic about life.”* (12 months)	*“She (...) has an amazing zest for life. She does everything with such enthusiasm and joy”*	No change
c. Mischief and sassiness	*“Sleeping less, more mischievous”*	*“A very happy puppy who is playful loving brave and sassy, a little naughty but a joy to own”* (6 months)	*“[Dog’s name] is a playful, cheeky pickle.”* (12- months)	*“Her eyebrows raise when I ask her to do something and she can be very clever/sneaky in trying to get up to mischief!”*	Dogs described as mischievous at all timepoints; more common in the 6- and 9-month surveys.
d. Unique character	*“Pushes new boundaries each day. Has character”*	*“We want a dog with real character and that means sometimes mischievous.”* (6 months)	*“[Dog’s name] is a real character and a joy to share the house with.”* (18 months)	*“He still pinches my pants and runs round the garden with them!—he is such a Scamp and just great fun—a real character”*	Dog’s unique characteristic discussed at all timepoints. From 6 months onwards, comments about dogs ‘becoming’ or ‘growing into’ themselves and their unique character are more noticeable.
e. Affection	*“Very kissy. Loves cuddles”.*	*“She (…) loves nothing more than a cuddle”.* (9 months)	*“[Dog’s name] has developed into a lovely* *affectionate dog*” (12- months)	*“The cuddles we have in the morning, before we start our day. It is a lovely calm moment in our day and a great* *beginning”*	Dogs described as affectionate at all timepoints; more common in the 18-month and 2-year surveys.
f. Calmness and ability to relax	*“[Dog’s name] is the calmest puppy I have ever known.”*	*“I appear to have been lucky in that she seems to have a very loving laid back personality where little bothers her”* (6 months)	*“[Dog’s name] has calmed down a lot since a puppy.”* (12 months)	*“He is calm, relaxed and an angel around the home.”*	In the 12–16-week and 6-month surveys, dog’s calm/relaxed behaviour was seen as an exception. Owners commented on their dogs becoming calmer, more settled and more relaxed from 12 months onwards.
g. Curiosity	*“[dog’s name] is a (…), friendly and inquisitive puppy.”*	*“[Dog’s name] appears to be an alert, inquisitive and intelligent pup”* (6 months)	*“He is very bright and curious and good natured”* (18 months	*“He is a very curious dog who always likes to check out everything new in his environment, sniffing around at everything.”*	No change
h. Confidence	*“Although [dog’s name] was very quiet when we first picked her up, she certainly has grown in confidence and has a huge fun character”*	*“She loves her older two sisters and is confident and happy.”* (9 months)	*“Very active and confident”* (18 months)	*“He is also a brave dog who might be a little tentative about something at first, but then will always end up achieving* *whatever it is he wants.”*	Comments on dog’s growth in confidence were common in the 12–16-week survey and continued until 18 months. Concerns regarding dog’s lack of confidence were raised in all but 6-month surveys. In the 12- and 18-month surveys, many owners stated that dog’s confidence is improving.
3. Training					
a. Training progress	*“His toilet training is amazing I’m so proud of him and how fast he is learning.”*	*“He is learning to be walked off lead and his recall is incredible! Very proud puppy parent!”* (6 months)	*“[Dog’s name] is very easy to train and learn new tricks”* (18 months)	*“A delightful dog with a lot of charm, easy to work with, tries to please”*	In 12–16-month surveys, comments on how quickly dog learns and the dog’s progress were common. Owners also noticed improvement in dog’s trainability from 12 months onwards.
b. Ability to fit with the family	*“She is travelling in the car well (…) and she is growing in confidence in new social situations, including going into shops, cafes and pubs. Fab!”*	*“He’s great (…) meeting new people, not fussed about being left on his own, has slept in lots of different houses and doesn’t seem phased, and has generally adjusted to life with us fantastically”* (6 months)	*“She has coped well with family changes/absences and the disruptions this caused to her routines.”* (18 months)	*“She adapts well to new situations, I feel like I can take her anywhere!”*	In all surveys, owners noticed how quickly their dog adapted to the family (early surveys) or to changes within family (later surveys).
4. Physical appearance and healthy development					
	*“[Dog’s name] is a wonderful border collie puppy, who is developing beautifully.”*	*“[Dog’s name] in developing into a big, beautiful, healthy dog”* (6 months)	*“He is a strong, muscular ‘tigger’”* (18 months)	*“She has very expressive facial expressions, and makes me laugh all the time.”*	Owners commented on dog’s cuteness and growth more often in the 12–16-week surveys.

**Table 4 animals-13-01863-t004:** Sub-themes and codes that contribute to ‘Negative experiences of dog ownership’ meta-theme at different timepoints. Quotes that illustrate a particular sub-theme/code are provided in addition to comments regarding the direction of change.

Code	12–16 Weeks	6 and 9 Months	12 and 18 Months	2 Years	Change over Time
1. Settling into home and family life					
a. Ability to settle and adapt	*“We are finding it hard during the night. He sometimes wakes at 1–2 am for a wee. (…). We take it in turns to get up early and then that person has an afternoon nap! We just don’t know how* *to get over this. (…)”*	*“Sometimes he will not stay in his bed at night and barks until we get up and let him into our room. He won’t settle unless he is with us. I would say this is 2–3 nights a week for the past month.”* (6 months)	*“Difficulty settling in different places, including on long car journeys (fine on short trips).”* (12 months)	*“He doesn’t like things to change, and gets very upset if things move place or aren’t in the right routine.”*	More common in the 12–16-week survey than later. In the early survey, issues with settling in were more often related to dog settling in at night. Later, issues with settling in included dog struggling to adapt to change in their routine.
b. Difficult to leave alone	*“Difficult to leave alone. Follows me and needs to be near someone all the time.”*	*“He is also extremely bonded to me (…) This causes issues if we’re out and I pop into a shop or go out of sight. He will become anxious and tremble.”* (6 months)	*“I am unable to leave him at all. He barks and whines when on his own.”* (12 months)	*“Doesn’t like to be on her own.”*	No change
c. Destroys things	*“Picking up stones of all sizes and chewing them, sometimes swallowing them. Chewing plastic and paper and swallowing some of it.”*	*“(…) he stills chews shoes, socks and anything loose, but this is becoming less so sure he will stop at some point!”* (6 months)	*“(…) He still steals and destroys odd items if they are left out: yellow washing-up cloths, letters and magazines, plastic bags—we have learnt to be very tidy!”* (18 months)	*“He eats socks. Then forgets he’s hiding it in his mouth, then I have to go poopascoop socks* *from the garden!”*	Dog chewing inappropriate items and being destructive was discussed more often in the 12–16-week survey, when the behaviour was seen as expected due to young age or teething. From 6 months onwards, destructive behaviour was discussed as something that a dog was ‘still’ doing.
2. Issues with training					
a. Housetraining	*“We are finding house training a challenge. [Dog’s name] seems to pee a lot, we take him outside upwards of 15 times a day but still have accidents.”*	*“[S]eems to have gone a bit backwards with toilet training. He had stopped peeing/pooping in the house but now the weather has gotten bad he has started again.”* (9 months)	*“Will also occasionally wee indoors just after a walk outside. Also toilets indoors when staying with friend.”* (12 months)	*“He’s still weeing up things in the house!”*	Housetraining issues were discussed at all timepoints, albeit more frequently in the 12–16-week survey.
b. Ignoring commands/testing boundaries	*“Pushes new boundaries each day.”*	*“Recall has regressed, after lots of intensive training it seems he is challenging things and testing boundaries. Had to take training back to basics at times recently as interest to meet other dogs is greater than his desire to listen to the basic commands he knows and used to perform flawlessly a month* *or two ago.”* (6 months)	*“She does not come when called—often ignores me.”* (18 months)	*“How sassy she can be when she argues back if she starts barking and you tell her to be quiet she will bark back as if to be like no don’t tell me what to do.”*	Ignoring known commands, ‘pushing boundaries’ and, in particular, deterioration of dog’s recall was very common in comments in the 6-, 9- and 12-month surveys.
3. Challenging personality/dispositions					
a. Stubborn	*“He is very strongly self- willed.”*	*“[Dog’s name] is (…) very stubborn and is taking more training than we have had with other breeds.”* (9 months)	*“He is stubborn and I don’t think we’ll ever really get him not to pull on the lead.”* (18 months)	*“She can be very wilful. She will only do something because she wants to.”*	No change
b. Too excitable/boisterous	*“He’s very excitable (…) Jumping up at people whenever he sees them.”*	*“[Dog’s name] has started being very hyperactive and overexcited towards other dogs/people pulling and lunging at them on the walks with the intention of playing.”* (6 months)	*“gets wayyyyyy overexcited and overstimulated and finds it hard to focus on training or what he’s being asked.”* (12 months)	*“He still gets over aroused easily, especially around other dogs. He thinks they all want to play and his play is typical Doodle- very excited.”*	Discussed at all timepoints. In 6-month and later surveys, owners more often discussed this as ‘still’ being an issue, whereas in the 12–16-week survey, it was something that was expected.
c. Lack of confidence/nervousness	*“She is a little nervous at new situations and around other puppies at times.”*	*“We are a bit upset at how submissive she is. She seems to have little confidence. She is to be a gun dog and already my husband tells me to rehome her as she is too soft.”* (6 months)	*“Shyness with people she doesn’t know- more when she is indoors than outside.”* (18 months)	*“She is a more nervous dog than I would have hoped.”*	Nervousness and lack of confidence were mentioned most often in the 12–16-week survey.
d. Attention seeking/jealous	*“Barking at people, dogs and for attention.(…) shows signs of jealousy with our other* *dog.”*	*“Her jealousy of all attention that I give to (..) her sister. [Dog’s name] wants all attention from me focused on her only.”* (6 months)	*“Attention seeking barking.”* (18 months)	*“He’s pretty wimpy and needy! (…) He is very green eyed and doesn’t really like our other dogs being fussed, he tries to push his way in.”*	No change
4. Challenging behaviour					
a. Chasing animals	*-*	*“He ignores me on walks and has chased sheep when his* *longline snapped.”* (9 months)	*“Tends to chase livestock/deer if off lead so has to be kept on lead” much of the time in nearby deer park.”* (12 months)	*“She is a chaser and disappears in a flash when she catches an interesting smell. Chases birds, deer and twice she has chased sheep.”*	Not mentioned in the 12–16-week survey. Comments about chasing other animals most common in the 18-month survey.
b. Issues aroura other dogs	*“Mildly reactive (alarm barking, backing away) toward strange dogs when encountered unexpectedly.”*	*“He snarls and barks [at other dogs], but is wagging his tail at the same time. Sometimes he will turn on my other dog as an outlet for his excitement. They sound as if they are having a serious fight, but don’t seem to hurt one another”* (9 months)	*“She sometimes plays with them [other dogs], but soon gets bored of them and she can be quite grumpy and growls & snaps at them when she’s had enough.”* (18 months)	*“He can be reactive to some other males.”*	Comments regarding dog’s reactivity and aggression towards other dogs were more common from 12 months onward. At 2 years, a number of owners stated that this behaviour was improving.
c. Behaviour around people	*“Barking, snarling and snapping (usually when we eat).”*	*“[Dog’s name] can react (barking, pulling away) from some people we meet.”* (6 months)	*“Won’t allow strangers to touch him, not generally a problem but could be in the future.”* (18 months)	*“Her terrier tendency to sometimes be snappy and grumpy.”*	Issues around known and unfamiliar people were not common; listed more often 6 months onwards.
d. Pulling on lead	*“Pulls on lead but working on her.”*	*“Pulling on the lead, which he never used to do.”* (6 months)	*“He is pulling on the lead constantly so I have to keep re training him which is annoying.”* (12 months)	*“He still pulls on the lead.”*	Discussed in all surveys. Many owners thought that the behaviour was worse than previously in the 6- and 9-month surveys.
e. Barking	*“Very vocal when playing with other dogs (…).”*	*“He has just started to bark at people going past, at shadows and a lot of other things which is annoying and we hope he will get less barky as he learns what is to bark at and what isn’t’.”* (9 months)	*“Barking for nothing and when we get home.”* (12 months)	*“Prolonged loud barking at any noise at all.”*	Barking was discussed more often from 6 months onwards.

## Data Availability

The data are not publicly available due to the ethical approval of participant informed consent that included ‘Generation Pup’ participants being informed that we will remove all personally identifiable information before sharing data with universities and/or research institutions.

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
