# Peer review of "“It’s Like Living with a Sassy Teenager!”: A Mixed-Methods Analysis of Owners’ Comments about Dogs between the Ages of 12 Weeks and 2 Years"

_animals, 2023, doi:10.3390/ani13111863_

Round 1
Reviewer 1 Report
There is only 117 dogs in the longitudinal part of the study, whereas it would have been the only interest of this kind of study.
This paper only states the obvious : owners interprete their dog's behavior, are less and less tolerant about it with time...
We cannot say if their statement is close to be true or not at all because no clinical data is given about their dog.
This paper is only descriptive and brings nothing much in the field of behavioral medicine.
Author Response
Thank you for taking the time to read and review this manuscript. We addressed the comments below. Unless specified otherwise, when referring to line numbers below, we mean numbers seen when the ‘No markup’ option is selected in Word review mode.
There is only 117 dogs in the longitudinal part of the study, whereas it would have been the only interest of this kind of study.
We politely disagree with this comment. Although the dataset that includes a complete set of information about a dog across all timepoints is 117, 3577 comments about 1808 dogs (derived from different time points) were analysed. Comparison of themes and demographic data between the sample of 117 dogs and the whole population were made and no discernible differences were identified. Due to the paucity of information about dog owner experiences of living with dogs under the age of 2 [1] and young age being strongly associated with the risk of euthanasia and relinquishment [2], [3], we believe that research focused on owners’ experiences is of interest.
Moreover, this study draws on qualitative research methods. As such, one of the aims of this manuscript was to provide a qualitative understanding of dog ownership experiences during the first 2 years of a dog’s life and to compare findings of the textual and qualitative thematic analyses. Qualitative analysis is not intended to quantify themes but to show the range of experiences or patterns [4]. Within the qualitative approach, sample size does not reflect on the study’s rigour, rigour is instead based on the transparency of the analysis process, and incorporating quotes and indeed descriptions to illustrate the constructed themes [5].
This paper only states the obvious : owners interprete their dog's behavior, are less and less tolerant about it with time...
We politely disagree with this comment. It may be common sense, but to the best of our knowledge, this information has not been previously published using longitudinal data. We also characterise a number of positive and negative experiences that owners have with their dogs during the first 2 years of life. Information on this subject can be incorporated into human behaviour change interventions, which rely on a thorough understanding of owner motivation for dog ownership. We illustrate how owners’ sentiment change over time and show that some owners find their dog behaviour during puppyhood challenging. This information can be used to manage the owner’s expectations. We also characterise how owners explain dog behaviour and how these explanations change over time. Again, understanding of owner's perception of dog behaviour is crucial to facilitate educational and behaviour change interventions.
We cannot say if their statement is close to be true or not at all because no clinical data is given about their dog.
This is true. However, qualitative analysis never aims to uncover the truth. Instead, the goal is usually to characterise the patterns of perceptions or beliefs. The aims of this study were stated (in the original manuscript, lines 95-96) and did not include verifying whether owner perceptions are true or not. We have provided further explanation of the general aims of qualitative analysis (lines 200-208) and re-phrased study objectives (lines 86-92).
This paper is only descriptive and brings nothing much in the field of behavioral medicine.
We disagree. The manuscript employs a mixed methods approach, including qualitative thematic analysis and textual analysis. In qualitative thematic analysis, the provision of quotes and descriptions is necessary and indeed listed among the reporting standards [4], [5]. Behavioural medicine involves communication with dog owners and understanding of challenges they face, which this study outlines. Behavioural medicine additionally requires excellent communication with clients and the ability to educate about dog behaviour. An understanding of common frameworks for explaining dog behaviour and perceptions/ interpretations of behaviour that are inaccurate can help with this.
[1] A. G. Costa, T. Nielsen, R. Christley, and S. Hazel, “Highlighting gaps in puppy research using Bronfenbrenner’s bioecological theory of human development,” Human-Animal Interact., no. 2023, 2023.
[2] C. Pegram et al., “Proportion and risk factors for death by euthanasia in dogs in the UK,” Sci. Rep., vol. 11, no. 9145, 2021.
[3] E. Weiss et al., “Large Dog Relinquishment to Two Municipal Facilities in New York City and Washington, D.C.: Identifying Targets for Intervention,” Anim. 2014, Vol. 4, Pages 409-433, vol. 4, no. 3, pp. 409–433, Jul. 2014, doi: 10.3390/ANI4030409.
[4] V. Clarke, V. Braun, and N. Hayfield, “Thematic analysis,” Qual. Psychol. A Pract. Guid. to Res. methods, vol. 3, pp. 222–248, 2015.
[5] A. Tong, P. Sainsbury, and J. Craig, “Consolidated criteria for reporting qualitative research (COREQ): a 32-item checklist for interviews and focus groups,” Int. J. Qual. Heal. care, vol. 19, no. 6, pp. 349–357, 2007.
Reviewer 2 Report
Congratulations on addressing an area of research of interest and importance to the dog-owning public.
I only have a few minor comments and questions:
-Line 95. I believe this should be 'complement'
-Lines 134 & 213. Some confusion. Line 134 states "one dog from each multi-dog household was randomly excluded" whilst Line 213 states "After removing (at random) all dogs bar one from multi-dog households". Was only one dog per multi-dog household kept or excluded?
- Section 3.2. I am not sure how useful the word frequency is as so much is reliant on context. e.g. "get" is common at all time points. In what context was "get" used? Likewise, the two most common words at 9 months were "week" and "get". Again, context would be useful to know, perhaps giving examples.
-Line 274. From my understanding of Fig. 3a, analysis using Bing produced a mean sentiment that was negative for all time points but you state "median sentiment was neutral and mean sentiment was positive". I also couldn't see median sentiment in Fig 3.
-Line 291. Up to this point I had assumed 'love' to be 'love'. Here you state it is stem for 'lovely'. These words have totally different meanings. Which is correct?
-Line 398. In the sample description, were you able to see how many dog owners had previously owned dogs? If so, did you analyse whether this affected the results?
-Line 604. You state "Most attributions of dog behaviour referred to a dog's unmodifiable characteristics...". Is it possible to quantify this in the results?
Author Response
Thank you for taking the time to read and review this manuscript and for the suggestions on how to improve it. We explain how the comments were addressed below. Unless specified otherwise, when referring to line numbers below, we mean numbers seen when ‘No markup’ option is selected in Word review mode.
-Line 95. I believe this should be 'complement'
We re-worded the study objective (lines 85-92) and this word was removed. Study objectives now read: To explore this further, a better understanding of owner perceptions of dog behaviour is needed. Therefore the second objective of this study is to examine owners' perceptions and attributions of dog behaviour as dogs mature between the age of 12 weeks and 2 years. To this end, the study utilises free text responses from a longitudinal survey-based study, including responses to “any other information” question posed at the end of surveys at different timepoints. Both qualitative reflexive thematic analysis and quantitative textual analysis are used, and the third objective is to compare the two analytical approaches for the analysis of open-ended survey questions.
-Lines 134 & 213. Some confusion. Line 134 states "one dog from each multi-dog household was randomly excluded" whilst Line 213 states "After removing (at random) all dogs bar one from multi-dog households". Was only one dog per multi-dog household kept or excluded?
Thank you for spotting this discrepancy- we confirm that only one dog per household was included in the analysis. The text now reads: Ahead of analyses, one dog from each multi-dog household was randomly included to avoid household-level clustering. (Line 148).
- Section 3.2. I am not sure how useful the word frequency is as so much is reliant on context. e.g. "get" is common at all time points. In what context was "get" used? Likewise, the two most common words at 9 months were "week" and "get". Again, context would be useful to know, perhaps giving examples.
We agree that the findings need to be interpreted contextually. As “Dog’s love and bond with a dog” were identified as one of the main themes describing the positive experience of dog ownership, we hope that the context in which the most common words identified in Figure 1 occur is explained (e.g. discussion in lines 489-492). Retaining Figure 1 enables us to show how often owners used words derived from love in their comments, which is not as obvious from Figures 2 and 4.
We added a comment on the limitations of word frequency graphs (lines 551-555): “This suggests that whilst word frequency graphs may be useful to highlight macro-trends in word use over time, they need to be interpreted cautiously. Moreover, as the algorithm reduces words to stems to enable comparison, without context, it is unclear whether the high frequency of stem ‘love’ is due to reporting dogs as loved, talking about love for a dog or describing them as lovely.).
We also edited the results section (lines: 248-251) Aside from the 12-16 and 9-month weeks, at all timepoints words related to ‘love’ were most frequent (Figure 1). At 12-16 weeks the most common words were ‘bark’, ‘bite’, ‘jump’, and ‘play’. At 9-months, words related to ’love’ were relatively frequent, but the most often used words were common words, such as ‘get’ or ‘week’ (Figure 1).
-Line 274. From my understanding of Fig. 3a, analysis using Bing produced a mean sentiment that was negative for all time points but you state "median sentiment was neutral and mean sentiment was positive". I also couldn't see median sentiment in Fig 3.
Thank you for pointing this out. We clarified in the methods section that Figure 3a reflects the net sentiment rather than mean sentiment (as sentiments are expressed on different scales, calculation of a net sentiment i.e. a number of words coded as positive sentiment- negative sentiment enablers comparison). The text reads: Sentiment scores from individual tokens were added up for each survey response and to compare different lexicon dictionaries, the net sentiment (positive-negative) was plotted (lines 196-198).
-Line 291. Up to this point I had assumed 'love' to be 'love'. Here you state it is stem for 'lovely'. These words have totally different meanings. Which is correct?
The stemming algorithm works by conflating words with the same meaning rather than words that just have common linguistic roots (e.g. awe and awful have different stems). For this reason, stemming reduces some words in a non-obvious way and some stems do not correspond to linguistic word-stems- e.g. “happy” and “happiness” are stemmed as “happi” to differentiate from stems of words related to “happening” or “happened” (which stem to “happen”). ‘Love’ as a word stem comes up as a word that frequently contributes to the positive sentiment because of the high frequency of words like lovely, but also loving, loved and love itself. We added this to the limitations of this approach: “Moreover, as the algorithm reduces words to stems to enable comparison, without context, it is unclear whether the high frequency of stem ‘love’ is due to reporting dogs as loved, talking about love for a dog or describing them as lovely.)” (Line 553-555)
-Line 398. In the sample description, were you able to see how many dog owners had previously owned dogs? If so, did you analyse whether this affected the results?
Yes, we did explore it, however, no patterns with respect to dog owner demographics were identified.
-Line 604. You state "Most attributions of dog behaviour referred to a dog's unmodifiable characteristics...". Is it possible to quantify this in the results?
Thank you for this suggestion. This finding is based on qualitative analysis. It is not appropriate to quantify qualitative findings. Instead, the use of semi-quantifications, such as many/ majority/ small numbers etc is common. The rationale behind this is that the use of numbers tends to detract from the objective of qualitative research which is to provide a range of opinions/ views and experiences or to characterise patterns [1]. Moreover, presenting numbers implies an ‘objective’ and measurable reality that is incompatible with the goals of qualitative research, which instead is focused on multiple and sometimes contradictory perspectives [2]. Because of the nature of qualitative data, reporting frequencies or percentages can be somewhat deceptive because respondents have not been given the same number of options to select from, as in quantitative survey questions. Numbers can also sometimes lead the reader to make generalisations about the findings, albeit unconsciously [1], [2] - in qualitative research views expressed less often are not necessarily less important than those shared by the majority as they help to develop a more nuanced understanding of the issue [3]. We clarified this passage of text to read: “Most common pattern of dog behaviour attribution referred to a dog’s unmodifiable characteristics, such breed, genetics or dog personality. “(line 587)
[1] J. Ritchie, J. Lewis, and G. Elam, “Designing and selecting samples,” Qual. Res. methods, pp. 77–108, 2003.
[2] J. A. Maxwell, “Using numbers in qualitative research,” Qual. Inq., vol. 16, no. 6, pp. 475–482, 2010.
[3] V. Braun and V. Clarke, “Using thematic analysis in psychology,” Qual. Res. Psychol., vol. 3, no. 2, pp. 77–101, 2006.
Reviewer 3 Report
This study describes people’s responses to the question ‘any other information’, analyzing information using a mixed methods approach. The paper addresses an interesting area, but I am concerned about the methods adopted to collect the data – I’m not entirely sure dog owners really knew what they were providing information on, or for. As such, I am not entirely sure that the study achieves its aims. My specific comments are outlined below.
Title: This is quite misleading – this study did not really explore owners’ perceptions or experiences. Owners simply wrote whatever they wanted in response to the question ‘any other information’
Abstract
Line 26. I would be inclined to change the word ‘mischievous’ to someone else as it implies a perception on the part of the owner and is rather subjective.
Methods
2.1. More detail is needed on Generation Pup. For example, how do get targeted for recruitment – have they had to acquire a dog from DogsTrust or from anywhere? Are there any inclusion/exclusion criteria for participation? Do participants have to provide consent to take part?
Likewise, whilst the authors are referring to another paper in relation to the methods, a bit of detail is needed on the survey itself. How many sections had the survey, what types of question were asked, etc. The details provided here are currently extremely vague, indeed almost completely lacking. Were any quantitative questions asked at all?
Table 1. It is unclear why the question ‘please describe the behaviour that you find to be a problem’ suddenly disappears when the dog is 2 years and becomes ‘The best thing about my dog is..’
Lines 137-138. It is unclear why the authors overlooked data specific to the nature of the problem behaviours owners reported on? Surely owners are less likely to go on to provide information on ‘any other information’ and the nature of the information they could provide here is extremely open-ended? Further, given that this question follows on from the problem behaviour question, are owners with ‘problem pets’ not going to be more biased to provide further details on their pet’s problematic behaviour? From the responses to the questions in Table 1, how many of these were actually responding to ‘any other information’ and how many to ‘please describe the behaviour you find to be a problem’? As currently written, it seems that the data for this study are simply based on people’s response to the question ‘any other information’? Surely not? In general, much more clarity is needed on what data were actually collected, by who, how many, etc. Although, even with this, I think the study is fundamentally flawed from a methododological perspective. It’s a shame, as with the right sorts of question and methodological approach at the outset, this type of information would be very useful.
A few minor mistakes scattered throughout the paper.
Author Response
Thank you for taking the time to read this manuscript and for the constructive criticism. Below we explain how we addressed the specific comments and explain further changes made to the manuscript prompted by the review process. Unless specified otherwise, when referring to line numbers below, we mean numbers seen when ‘No markup’ option is selected in Word review mode.
Title: This is quite misleading – this study did not really explore owners’ perceptions or experiences. Owners simply wrote whatever they wanted in response to the question ‘any other information’
We modified the title to read: “It’s Like Living With a Sassy Teenager!”: A Mixed-Methods Analysis of Owners’ Comments About Dogs Between the Ages of 12-Weeks and 2 Years.
Line 26. I would be inclined to change the word ‘mischievous’ to someone else as it implies a perception on the part of the owner and is rather subjective.
We politely disagree with this suggestion. The word mischievous was explicitly and frequently used by owners when explaining why their dog behaves in a particular way- the study was focused on subjective perceptions. We changed the wording to make it clearer (line 26): Problematic behaviours of young dogs were seen as “mischievous", unintentional and context-specific. Similar behaviours shown by older dogs were described as “deliberate”. In line with this suggestion we also changed the text in line 41: When dogs were young, owners described problematic behaviours as “mischievous”, unintentional and context-specific. Similar behaviours shown by older dogs were seen as “deliberate”.
2.1. More detail is needed on Generation Pup. For example, how do get targeted for recruitment – have they had to acquire a dog from DogsTrust or from anywhere? Are there any inclusion/exclusion criteria for participation? Do participants have to provide consent to take part?
Thank you for this suggestion. The inclusion criteria were provided (lines 106-109). The statement regarding consent is in line 661, as per journal requirements (as a statement about consent). We included text regarding study advertisement and recruitment so that the whole section now reads: ‘Generation Pup’ is open to participants who are residents of the United Kingdom (UK) or the Republic of Ireland (ROI); aged 16 years or over, and who own a puppy of any breed, mix-breed or cross. The study does not specify exclusion criteria related to the way dogs were acquired, as long as at the time of registration a dog is younger than 16 weeks of age (or younger than 21 weeks if a puppy entered the UK/ROI through quarantine). Study recruitment is ongoing and will pause when 10,000 dogs are recruited. Participants were recruited through social media, radio interviews, advertisement at veterinary practices, dog training venues and articles in veterinary, dog-related and other publications (lines 104-112)
Likewise, whilst the authors are referring to another paper in relation to the methods, a bit of detail is needed on the survey itself. How many sections had the survey, what types of question were asked, etc. The details provided here are currently extremely vague, indeed almost completely lacking. Were any quantitative questions asked at all?
Thank you for this suggestion. We opted for a succinct description, as information about the survey content has previously been published in an open-access publication. We provided further information in lines 112-123 which reads: “Each survey has between 2-19 sections. The introductory surveys completed as a part of registration (1-3 weeks after acquisition or until 16-weeks of age), collect information about the owner, household, puppy, and puppy acquisition. Topics covered in the later surveys analysed here include: introducing puppy to household, puppy’s/dog’s experiences, activities, meeting other people, meeting dogs, puppy’s/ dog’s behaviour, puppy’s/dog’s day, puppy’s sleep, diet, training approaches, health, surgery, neutering, insurance, dog’s boarding/kennelling experience, breeding, exercise, mobility, reflection and other information. Topics are repeated at regular intervals; topics related to dog behaviour, day, health, other information were included in all surveys analysed here (please see [1] for details). Each survey includes primarily close-ended questions and free text boxes that enable owners to expand, clarify or provide an alternative response to the close-ended questions. Open-ended questions included in the analysis are specified in Table 1.” Quantitative questions were asked, and manuscripts that draw on quantitative data are in preparation.
Table 1. It is unclear why the question ‘please describe the behaviour that you find to be a problem’ suddenly disappears when the dog is 2 years and becomes ‘The best thing about my dog is..’
Thank you for pointing this out. We added a rationale for question selection (lines 125-136):
“It has been suggested that open-ended questions within a survey can be used to explore general experiences and reflections related to other topics covered in a survey [2]. For this reason, the “any other information” question was selected from all but the last timepoint of interest. This question was optional. Questions about the best, the funniest and the most annoying thing about one’s dog, asked only in the 2-year survey, were included as they explicitly enquire about owner’s experiences and enable insight into owners’ perceptions of dog behaviour. The preliminary reading of responses to the “any other information” question highlighted frequent comments related to the things owners enjoyed about their dogs. Therefore, the free-text question about behaviours that owners found to be a problem was included to learn more about behaviours owners find challenging and to explore the perception of these behaviours. Owners were asked to answer this question if they specified that their dog shows a behaviour they find to be a problem.”
Lines 137-138. It is unclear why the authors overlooked data specific to the nature of the problem behaviours owners reported on? Surely owners are less likely to go on to provide information on ‘any other information’ and the nature of the information they could provide here is extremely open-ended? Further, given that this question follows on from the problem behaviour question, are owners with ‘problem pets’ not going to be more biased to provide further details on their pet’s problematic behaviour? From the responses to the questions in Table 1, how many of these were actually responding to ‘any other information’ and how many to ‘please describe the behaviour you find to be a problem’? As currently written, it seems that the data for this study are simply based on people’s response to the question ‘any other information’? Surely not? In general, much more clarity is needed on what data were actually collected, by who, how many, etc. Although, even with this, I think the study is fundamentally flawed from a methododological perspective. It’s a shame, as with the right sorts of question and methodological approach at the outset, this type of information would be very useful.
Thank you for these suggestions.
- We did not overlook the question related to behaviour problem or its nature, analysis of problem behaviours was not among the objectives of this study (before corrections, listed in lines: 95-99, now re-phrased in lines: 79-80; 86-92).
- As stated in the introduction, some of this analysis has already been conducted and further work is in preparation (lines: 91-93 in the original submission). We do not think it is appropriate to include this work in the current publication, as studies about problem behaviour based on Generation Pup data follow completely different methodologies, use different datasets (different time points) and ultimately, are on a different subject to this study.
- Owners who report problem behaviour may be more biased towards sharing further information. We acknowledged the reporting bias in the study limitations section. We have now further elaborated on this point in lines 567-570: The “any other comments” question was not compulsory and the question about problem behaviour was asked if a problem was first reported. It is possible that only owners who felt particularly strongly about their experiences offered comments, biasing the findings. We also highlighted this limitation in the study abstract (lines 28-29 and 44-45).
- Information on number of responses to “Any other information” and “Problem behaviour” questions was added to Table 1. To make the limitations of this study explicit, we changed from the number of all comments analysed to number of answered questions in Table 1. We also clarified there which questions were included in textual analysis.
- We politely disagree that the study is fundamentally flawed from a methodological perspective. It is a mixed-method study, majority of the analysis follows a qualitative paradigm. It was not our aim to quantify the responses or apply statistical analysis to this data. We are currently working on other publications that follow a quantitative approach. This study aimed to report on the perceptions of behaviours and owner experiences reported in free-text survey responses to the survey. During the review process, we additionally added a new objective of evaluating the strengths and limitations of this data. As previously suggested, free-text survey responses are valuable but difficult to analyse and under-utilised resources [2]. The two analytical approaches yielded complementary findings. We recognised the limitations of this study and in the course of this review, we strived to make them more visible to the reader, as outlined above.
A few minor mistakes scattered throughout the paper.
The manuscript was proofread and identified mistakes were corrected.
[1] J. K. Murray et al., “‘Generation Pup’ – protocol for a longitudinal study of dog behaviour and health,” BMC Vet. Res., vol. 17, no. 1, 2021, doi: 10.1186/s12917-020-02730-8.
[2] A. O’Cathain and K. J. Thomas, “‘ Any other comments?’ Open questions on questionnaires–a bane or a bonus to research?,” BMC Med. Res. Methodol., vol. 4, no. 1, pp. 1–7, 2004.
Round 2
Reviewer 1 Report
The authors did a huge effort to improve the quality of this paper.
I still find that this does not bring much to the field, but since the journal seems to be willing to publish this paper I am not against it.
Reviewer 3 Report
The authors have addressed all of my comments and concerns with good attention to detail.